# OpenReview forum: "More is Less: The Pitfalls of Multi-Model Synthetic Preference Data in DPO Safety Alignment"
_colmweb.org/COLM/2025/Conference — COLM 2025_

### Official Review · Reviewer_5pKC · 2025-04-26

**Rating:** 6
**Confidence:** 3
**Ethics Flag:** 1

**Summary:**

This paper is on safety alignment via Direct Preference Optimization (DPO) using synthetic preference data. The key message of this paper is a critical phenomenon in the DPO alignment. Specifically, although multi-model generated data enhances performance on general tasks by providing diverse responses, it also tends to facilitate reward hacking during training. The authors also observed that multi-model dataset leads to a high attack success rate (ASR) against jailbreaking prompts. Through extensive experiments, the authors conclude that single-model generation consistently outperforms multi-model methods in preventing jailbreaking attacks.

**Questions To Authors:**

[Q1] Are the authors' main claims true for the LLMs that are aligned with RLHF or other direct methods (e.g., IPO, SimPO)?

[Q2] Do the authors think that DPO is particularly vulnerable to the phenomenon the authors observed?

[Q3] Does reward hacking discussed in 3.4 results only from data generation process? Does this phenomenon occur for other DPO parameters, such as $\beta$ or learning rate?

**Reasons To Accept:**

1. This paper is well-written and organized. I think the key contributions are clear and the readers can easily understand the main claims of this paper.

2. The authors have conducted their experiments using several dataset generation approaches (as written in lines 151-170) and models (as shown in Figure 1). I feel the authors try to obtain a general knowledge about DPO alignment.

3. The final message (in lines 334-337) is useful in the research and development for LLM safety.

> We demonstrate that single-model generation, where models use their own generated responses for alignment, consistently outperforms multi-model methods that incorporate responses from external sources like humans or other agents, in preventing jailbreaking attacks.

**Reasons To Reject:**

I have several concerns regarding this paper.

1. While I do understand that DPO is a widely-used algorithm for LLM alignment (or even other foundation models), it is unclear whether the key messages of this paper are true with other alignment methods (e.g., RLHF or IPO). If this is a specific phenomenon of DPO, then it would be better what is the issue of DPO or how we can mitigate it.

2. While the discussion in lines 243-258 is interesting, I am not fully convinced whether data generation approach is the only reason of reward hacking. I agree with the authors that GPT40+Self provides drastic preference labels, but it is quite natural for this data generation approach to lead to overfitting with the same $\beta$ or learning rate. I am not fully convinced whether all the responsibility can be placed on the data generation approaches. Is there any possibility that the authors set unreasonable parameters for such datasets?

3. There are missing citations regarding safety-alignment using DPO.

- Wachi et al. "Stepwise alignment for constrained language model policy optimization." Advances in Neural Information Processing Systems 37 (2024): 104471-104520.
- Huang et al. "One-shot safety alignment for large language models via optimal dualization." Advances in Neural Information Processing Systems 37 (2024): 84350-84383.

### Minor comments
In Section 2, there are some dubious mathematics.
- What is $\pi_\text{rel}$? Is this $\pi_\text{ref}$.
- What is the difference between $\pi_\rho$ and $\pi_\theta$?
- While it may be clear for most of the readers, please define $\sigma(\cdot)$ as a sigmoid function.

---

> ### Author Response · Authors · 2025-06-01
>
> We appreciate the reviewer's thoughtful comments and the opportunity to clarify several points. Below, we address each concern:
>
> > **W1&Q1: Are the authors' main claims true for the LLMs that are aligned with RLHF or other direct methods (e.g., IPO, SimPO)?**
>
> We thank the reviewer for raising this important question and we address the two parts of this question separately:
>
> **1. For other direct preference optimization methods (e.g., IPO, SimPO):**
>
> Yes, our main claims extend to other direct preference optimization methods beyond DPO. We conducted IPO experiments on Llama3.1-8B, employing the same data generation strategies as in our DPO experiments. The results are as follows:
>
> | Method          | ASR      |
> | --------------- | -------- |
> | HC + Self       | 0.47     |
> | GPT4o + Self    | 0.19     |
> | Stronger + Self | 0.09     |
> | Human-Labeled   | 0.26     |
> | Self + RM       | **0.01** |
>
> Consistent with DPO, **Self + RM** achieves the lowest ASR (0.01), while preference data from stronger models or humans leads to significantly higher ASRs. This confirms that our core conclusion—**self-generated preference data filtered by an RM yields the most robust alignment**—also holds for IPO.
>
> Furthermore, methods like SimPO introduce adjustments such as length-normalized reward scoring to address length bias, as discussed in the Discussion section of our paper. While such methods may help mitigate specific artifacts (e.g., response length), they **do not fully resolve the vulnerabilities** tied to multi-model preference data. As shown in Table 2 and Figure 3, length control alone is insufficient to improve safety when using externally sourced preferences.
>
> **2. For RLHF methods** :
> We do not analyze RLHF methods (e.g., PPO, GRPO) in this work because they do not allow for clean isolation of the effects of different preference data sources. Unlike direct preference optimization methods such as DPO, which operate offline on fixed preference datasets, RLHF involves on-policy rollouts where the model generates its own responses and updates based on reward model scores. While the reward model in RLHF may be trained using different data sources, the policy is always optimized using rewards assigned to self-generated rollouts, making it difficult to disentangle the influence of the data used to train the reward model from that of the policy's own outputs. Our focus is on direct methods like DPO, where the setup enables controlled comparisons of data generation strategies and their impact on safety alignment.
>
>
> > **Q2: Do the authors think that DPO is particularly vulnerable to the phenomenon the authors observed?**
>
> Yes, we think DPO and its variants are indeed more vulnerable when the preference data come from multiple, heterogeneous models. Unlike online RL methods (e.g., PPO) that continuously generate new rollouts (self-generated responses), DPO is trained on the fixed preference dataset. Any stylistic artifacts that differentiate one source model’s answers from another’s are therefore “baked in,” giving the policy stable shortcuts to exploit. Our work highlights the need to scrutinize multi-model preference data when using direct preference optimization methods.
>
>
> > **W2&Q3: Does reward hacking discussed in 3.4 results only from data generation process? Does this phenomenon occur for other DPO parameters, such as β or learning rate?**
>
> To isolate the cause of reward hacking, we ran controlled experiments on the GPT-4o + Self dataset using Llama3.1-8B, varying $\beta$ from 0.01 to 0.9 and testing two learning rates: 5e-7 and 1e-7.
>
> As shown in the table below, ASR remained consistently high and stable across all settings. This suggests that hyperparameter tuning does not alleviate reward hacking, reinforcing our conclusion that the issue mostly stems from the data.
>
> |           | $\beta=0.01$ | $\beta=0.1$ | $\beta=0.2$ | $\beta=0.3$ | $\beta=0.4$ | $\beta=0.5$ | $\beta=0.6$ | $\beta=0.7$ | $\beta=0.8$ | $\beta=0.9$ |
> | --------- | ------------ | ----------- | ----------- | ----------- | ----------- | ----------- | ----------- | ----------- | ----------- | ----------- |
> | $lr=1e-7$ | 0.54         | 0.54        | 0.56        | 0.56        | 0.53        | 0.53        | 0.56        | 0.56        | 0.56        | 0.57        |
> | $lr=5e-7$ | 0.55         | 0.56        | 0.55        | 0.56        | 0.55        | 0.56        | 0.54        | 0.53        | 0.54        | 0.56        |
>
>
> > **W4: Minor comments & missing citations**
>
> We appreciate the reminder of references and math typos. We will fix the typos ($\pi_{rel}$ is $\pi_{ref}$ and $\pi_\rho$ is $\pi_\theta$), describe sigmoid function $\sigma(\cdot)$ and add the following references in our updated version.
> - Wachi et al., “Stepwise alignment for constrained language model policy optimization.” NeurIPS 37, 2024.
> - Huang et al., “One-shot safety alignment for large language models via optimal dualization.” NeurIPS 37, 2024.

---

> ### Author Response · Authors · 2025-06-06
>
> With only a few days left in the discussion period, we would be most grateful if you could review our responses and share any further feedback at your convenience so that we may move closer to consensus on this paper.

---

> ### Comment · Reviewer_5pKC · 2025-06-08
> **Acknowledgement**
>
> Thank you for providing the authors' feedback. I have read other reviews and rebuttals. The responses mostly addressed my concerns, so I updated the score accordingly

---

### Official Review · Reviewer_1EjX · 2025-05-12

**Rating:** 6
**Confidence:** 3
**Ethics Flag:** 1

**Summary:**

This paper examines methods for generating preference alignment data. It shows that self-alignment—using a model’s own generated responses—leads to significantly stronger safety performance compared to multi-model synthetic preference data. The latter approach, while achieving similar general-task performance, is susceptible to reward hacking and demonstrates weaker robustness against jailbreak attacks.

**Reasons To Accept:**

1. This paper is well-organized and easy to understand.
2. The study systematically compares multiple preference data generation strategies across both general performance and safety alignment, using diverse models (LLaMA, Mistral, Qwen) and rigorous benchmarks like AdvBench.
3. This paper uncovers an important phenomenon—that multi-model synthetic preference data, despite its diversity, can degrade safety by enabling reward hacking, whereas single-model data yields safer outcomes.

**Reasons To Reject:**

1. The notion of "safety" in this study is predominantly operationalized in terms of jailbreak resistance. However, critical dimensions such as toxicity mitigation and fairness remain unexplored, which constrains the broader applicability of the findings.

2. The experimental results are primarily derived from 7B-scale models. The study does not investigate whether the observed patterns persist in significantly larger models, raising questions about the scalability validity of the conclusions.

---

> ### Author Response · Authors · 2025-06-01
>
> We thank the reviewers for their careful consideration. We greatly appreciate the positive comments and address major concerns below.
>
> > **W1: The notion of "safety" in this study is predominantly operationalized in terms of jailbreak resistance. However, critical dimensions such as toxicity mitigation and fairness remain unexplored, which constrains the broader applicability of the findings.**
>
> To further validate our findings, we evaluated toxicity mitigation and fairness using TOXIGEN [1], which covers over 135k toxic and 135k benign statements across 13 minority identity groups (e.g., African Americans, women, LGBTQ+ folks, etc.). We scored model outputs on a 1–5 Toxic-Score using GPT-4o. As shown in the below table, the toxic evaluation results from Llama3.1-8B align with our main conclusion—**Self+RM** achieves the lowest toxicity, while multi-model methods, such as using GPT-4o responses as chosen examples, result in much higher Toxic-Scores.
>
> These results indicate that Self+RM not only strengthens jailbreak resistance but also reduces toxic outputs, underscoring its broader effectiveness for safety, including toxicity mitigation and fairness considerations.
>
> | Method        | Toxic-Score   |
> | ------------- | :-----------: |
> | HC+Self       |     1.87      |
> | GPT4o+Self    |     2.17      |
> | Stronger+Self |     2.13      |
> | Peer+RM       |     1.85      |
> | All+RM        |     2.00      |
> | Human-Labeled |     1.88      |
> | **Self+RM**   |   **1.74**    |
>
> Reference
>
> [1] Hartvigsen, Thomas, et al. “Toxigen: A large-scale machine-generated dataset for adversarial and implicit hate speech detection.” ACL (2022).
>
> > **W2: The experimental results are primarily derived from 7B-scale models. The study does not investigate whether the observed patterns persist in significantly larger models, raising questions about the scalability validity of the conclusions.**
>
> We agree that it is valuable to investigate the generalizability of our findings for the larger models. Thus, we conducted experiments on larger models: Llama-2-13B and Qwen2.5-14B.
>
> As shown in the table below, both larger models demonstrate the same relative trend across data sourcing methods: **Self+RM consistently yields the lowest ASR, while HC+Self and GPT-4o / Stronger+Self produce higher ASR values.**
>
> These results suggest that our core findings on the relative safety effectiveness of different data curation strategies are robust across model scales, further strengthening our conclusion. We will include these larger model results in the updated version of the paper.
>
> | Model       | Method        |   ASR    |
> | ----------- | ------------- | :------: |
> | Llama-2-13B | HC+Self       |   0.46   |
> | Llama-2-13B | Stronger+Self |   0.32   |
> | Llama-2-13B | GPT4o+Self    |   0.23   |
> | Llama-2-13B | Self+RM       | **0.13** |
> | Qwen2.5-14B | HC+Self       |   0.47   |
> | Qwen2.5-14B | Stronger+Self |   0.55   |
> | Qwen2.5-14B | GPT4o+Self    |   0.26   |
> | Qwen2.5-14B | Self+RM       | **0.14** |

---

> > ### Comment · Reviewer_1EjX · 2025-06-08
> >
> > I appreciate your response and additional results, which address my key concerns. I have raised my score.

---

> ### Author Response · Authors · 2025-06-06
>
> With only a few days left in the discussion period, we would be most grateful if you could review our responses and share any further feedback at your convenience so that we may move closer to consensus on this paper.

---

### Official Review · Reviewer_HTMa · 2025-05-13

**Rating:** 7
**Confidence:** 3
**Ethics Flag:** 1

**Summary:**

This work explores how synthetic preference data generation affects LLM safety alignment, showing that self-generated data outperforms multi-model data, which leads to reward hacking and poor safety outcomes.

**Questions To Authors:**

1. In the first SFT stage, does it only use the winning samples for SFT?
2. In the online DPO settings, we still find that self-generated RM can significantly boost the performance. Does this phenomenon also apply to the online setting in terms of safety?

**Reasons To Accept:**

1. Identifies key challenges with multi-model synthetic data in safety alignment.
2. Provides thorough experimental evaluation across multiple benchmarks.
3. Offers novel insights into the effectiveness of self-generated data for safety.
4. The paper is easy to follow.

**Reasons To Reject:**

N/A

---

> ### Author Response · Authors · 2025-06-01
>
> We thank the reviewer for their positive and constructive feedback. We address the questions raised below:
>
> > **Q1: In the first SFT stage, does it only use the winning samples for SFT?**
>
> Yes, we follow the SPIN [1] paper setting and use only the winning samples for the first-stage SFT.
>
> [1] Chen, Zixiang, et al. "Self-play fine-tuning converts weak language models to strong language models." arXiv preprint arXiv:2401.01335 (2024).
>
>
> > **Q2: In the online DPO settings, we still find that self-generated RM can significantly boost the performance. Does this phenomenon also apply to the online setting in terms of safety?**
>
> Yes, our further experiments confirm that the benefits of self-generated preference data extend to the online DPO setting in terms of safety. We conducted experiments using Llama-3.1-8B under an online DPO setup for three iterations. In each iteration, we generated a 10k-sized preference dataset (self-gen + RM) using the current model. As shown in the table below, we found **self-gen + RM** continuously improves safety across iterations in the online DPO setting.
>
> | Iteration | KW-ASR   | GPT-ASR  | GPT-Score |
> | --------- | -------- | -------- | --------- |
> | 0 (sft)   | 0.82     | 0.56     | 3.62      |
> | 1         | 0.43     | 0.16     | 1.89      |
> | 2         | 0.35     | 0.09     | 1.43      |
> | 3         | **0.33** | **0.05** | **1.38**  |

---

> > ### Comment · Reviewer_HTMa · 2025-06-09
> > **Official Comments by Reviewer HTMa**
> >
> > Thank you for your response. I appreciate the additional experiments conducted in the online setting. This work provides valuable insights for the community, and I will maintain my original score.

---

### Official Review · Reviewer_r9Sn · 2025-05-14

**Rating:** 6
**Confidence:** 3
**Ethics Flag:** 1

**Summary:**

This paper investigates how different types of synthetic preference data affect the safety alignment of Large Language Models (LLMs) when using DPO. The key finding is that Multi-model generated preference data (where responses come from different models) performs poorly for safety alignment compared to single-model generated data (where the model being trained generates its own preference data).
Overall, this study offers valuable insights into preference data creation strategies in safe alignment, challenging the conventional view that outputs from more powerful models can serve as effective learning signals for less capable models.

**Reasons To Accept:**

1. This paper presents a counterintuitive but empirically well-validated finding: using self-generated data by the model for safe alignment is more effective than using data from stronger models or multi-model mixtures. This discovery holds broad practical significance for the community.
2. The authors conducted extensive experimental tests on six different models, including those in the Llama, Mistral, and Qwen series. The results consistently support its main argument, demonstrating the reliability of the research findings.
3. This paper is well-organized and well-written.

**Reasons To Reject:**

1. An important component in both general tasks and safety tasks experiments is the reward model, but this paper lacks a description of this component. Is the same reward model used in both experiments? If so, what is its source? Additionally, does the reward model perform at the same level across different types of tasks?
2. Although the authors tested the effectiveness of various model families, the sizes of these models were concentrated around 7 billion parameters. This may introduce a bias related to model scale in the conclusions. Would the conclusion hold for larger or smaller models?

---

> ### Author Response · Authors · 2025-06-01
>
> We thank the reviewers for acknowledging the strong conclusion of this work and the quality of the presentation. We address the concerns as follows:
>
> > **W1. An important component in both general tasks and safety tasks experiments is the reward model, but this paper lacks a description of this component. Is the same reward model used in both experiments? If so, what is its source? Additionally, does the reward model perform at the same level across different types of tasks?**
>
> We thank the reviewer for pointing out the omission of details about our reward model. We will include the following information in the revised version of the paper.
>
> In all our experiments, we use the same reward model: [**OpenAssistant/reward-model-deberta-v3-large-v2**](https://huggingface.co/OpenAssistant/reward-model-deberta-v3-large-v2). This model has been thoroughly evaluated in **RewardBench** (Lambert et al., 2024), a comprehensive benchmark designed to assess reward models across multiple dimensions, including general helpfulness, safety, and reasoning. We believe this model is a good choice for our study, as it offers a well-balanced performance across both general and safety-critical tasks.
>
> Overall Performance on RewardBench：
>
> | Reward Model                                   | Overall Score | Chat | Safety | Reasoning |
> | ---------------------------------------------- | ------------- | ---- | ------ | --------- |
> | OpenAssistant/reward-model-deberta-v3-large-v2 | 61.6          | 89.4 | 73.4   | 38.5      |
>
> Below, we provide detailed model’s performance on general and safety-specific evaluation subsets from RewardBench:
>
>  1. **General Task Benchmarks**
> Performance on two widely-used open-ended instruction-following benchmarks:
>
> | Reward Model                                   | AlpacaEval Easy ↑ | AlpacaEval Length Bias ↑ | AlpacaEval Hard ↑ | MT-Bench Easy ↑ | MT-Bench Medium ↑ |
> | ---------------------------------------------- | ----------------- | ------------------------ | ----------------- | --------------- | ----------------- |
> | OpenAssistant/reward-model-deberta-v3-large-v2 | 83.2              | 99.0                     | 41.1              | 96.8            | 100.0             |
>
>  2. **Safety Benchmarks**
> These benchmarks evaluate the model’s ability to (1) appropriately refuse harmful content and (2) avoid overly conservative refusals due to lexical similarity. Prompts and chosen/rejected pairs are adapted from modified versions of **XSTest** [Röttger et al., 2023] and **Do-Not-Answer** [Wang et al., 2023].
>
> | Reward Model                                   | Safety Avg. ↑ | Refuse Dangerous ↑ | Refuse Offensive ↑ | XSTest: Should Refuse ↑ | XSTest: Should Respond ↑ | Do Not Answer ↑ |
> | ---------------------------------------------- | ------------- | ------------------ | ------------------ | ----------------------- | ------------------------ | --------------- |
> | OpenAssistant/reward-model-deberta-v3-large-v2 | 75.1          | 82.0               | 83.0               | 76.5                    | 83.2                     | 40.4            |
>
> References
> - Lambert, Nathan, et al. "_RewardBench: Evaluating Reward Models for Language Modeling_." arXiv preprint [arXiv:2403.13787](https://arxiv.org/abs/2403.13787) (2024).
> - Röttger, Paul, et al. "_XSTest: A Test Suite for Identifying Exaggerated Safety Behaviors in Large Language Models_." arXiv preprint [arXiv:2308.01263](https://arxiv.org/abs/2308.01263) (2023).
> - Wang, Yuxia, et al. "_Do-Not-Answer: A Dataset for Evaluating Safeguards in LLMs_." arXiv preprint [arXiv:2308.13387](https://arxiv.org/abs/2308.13387) (2023).
>
>
> > **W2: Although the authors tested the effectiveness of various model families, the sizes of these models were concentrated around 7 billion parameters. This may introduce a bias related to model scale in the conclusions. Would the conclusion hold for larger or smaller models?**
>
> We agree that it is valuable to investigate the generalizability of our findings for the larger models. Thus, we conducted experiments on two larger models: Llama-2-13B and Qwen2.5-14B.
>
> As shown in the table below, both larger models demonstrate the same relative trend across data sourcing methods: **Self+RM consistently yields the lowest ASR, while HC+Self and GPT-4o / Stronger+Self produce higher ASR.**
>
> These results show that our core findings on the safety impact of data curation hold across model scales, further reinforcing our conclusion. We will include these larger model results in the updated paper.
>
> | Model       | Method        |   ASR    |
> | ----------- | ------------- | :------: |
> | Llama-2-13B | HC+Self       |   0.46   |
> | Llama-2-13B | Stronger+Self |   0.32   |
> | Llama-2-13B | GPT4o+Self    |   0.23   |
> | Llama-2-13B | Self+RM       | **0.13** |
> | Qwen2.5-14B | HC+Self       |   0.47   |
> | Qwen2.5-14B | Stronger+Self |   0.55   |
> | Qwen2.5-14B | GPT4o+Self    |   0.26   |
> | Qwen2.5-14B | Self+RM       | **0.14** |

---

> > ### Comment · Reviewer_r9Sn · 2025-06-09
> >
> > Thank you for your response. It would be beneficial to incorporate these results into the next version of the paper. I will maintain my score.

---

> ### Author Response · Authors · 2025-06-06
>
> With only a few days left in the discussion period, we would be most grateful if you could review our responses and share any further feedback at your convenience so that we may move closer to consensus on this paper.

---

### Decision · Program_Chairs · 2025-07-08

**Decision:**

Accept

**Comment:**

The paper studies the effect of using data from multiple different policy models in DPO on safety alignment (mostly, adversarial robustness). The authors compare a range of data generation strategies where the preferred responses are generated by GPT4o, stronger model, or best-of-6 responses from different models according to a Reward Model (RM). The authors first show that all methods achieve similar performance on some standard capability benchmarks. However, they show that all approaches that mix data across multiple source models lead to substantial degradation in adversarial robustness. Then, the authors hypothesize that  DPO with multiple policy models is susceptible to relying on spurious features, and reward hacking. The authors try to identify specific features (response length, format) and control for them, but the results are not very conclusive.

Several reviewers raised concerns about (1) the experiments all being done on small models, (2) the quality of the reward models, (3) hyper-parameter robustness of the results and (4) additional safety metrics beyond adversarial robustness. The authors provided additional experiments for all of these concerns, generally addressing them. All reviewers are recommending acceptance.

Overall, the paper presents a clean result, which is both somewhat surprising and useful.